# A Scoping Review of the Relationship between Intermittent Fasting and the Human Gut Microbiota: Current Knowledge and Future Directions

**DOI:** 10.3390/nu15092095

**Published:** 2023-04-26

**Authors:** Alina Delia Popa, Otilia Niță, Andreea Gherasim, Armand Iustinian Enache, Lavinia Caba, Laura Mihalache, Lidia Iuliana Arhire

**Affiliations:** Faculty of Medicine, University of Medicine and Pharmacy “Grigore T Popa”, 700115 Iasi, Romania; alina.popa@umfiasi.ro (A.D.P.); iustinian-armand.enache@umfiasi.ro (A.I.E.); lavinia.caba@umfiasi.ro (L.C.); laura.mihalache@umfiasi.ro (L.M.); lidia.graur@umfiasi.ro (L.I.A.)

**Keywords:** intermittent fasting, gut microbiota, metabolic syndrome, time-restricted feeding, Ramadan fasting

## Abstract

Intermittent fasting (IF) has been promoted as an alternative to dietary caloric restriction for the treatment of obesity. IF restricts the amount of food consumed and improves the metabolic balance by synchronizing it with the circadian rhythm. Dietary changes have a rapid effect on the gut microbiota, modulating the interaction between meal timing and host circadian rhythms. Our paper aims to review the relationships between IF and human gut microbiota. In this study, the primary area of focus was the effect of IF on the diversity and composition of gut microbiota and its relationship with weight loss and metabolomic alterations, which are particularly significant for metabolic syndrome characteristics. We discussed each of these findings according to the type of IF involved, i.e., time-restricted feeding, Ramadan fasting, alternate-day fasting, and the 5:2 diet. Favorable metabolic effects regarding the reciprocity between IF and gut microbiota changes have also been highlighted. In conclusion, IF may enhance metabolic health by modifying the gut microbiota. However additional research is required to draw definitive conclusions about this outcome because of the limited number and diverse designs of existing studies.

## 1. Introduction

Classical dietary caloric restriction (DCR) is not a sustainable method to treat overweight and obesity, which are becoming a global public health issue. The majority of patients lose weight within the first months of treatment; however, they regain weight when they stop following the recommendations. The drop-out rate can range from 20 to 40% [1], creating difficulties in the nutritional treatment of obesity. As a strategy to combat obesity, intermittent fasting (IF) has been proposed as an alternative to DCR, although its efficacy is still debatable. However, different studies show that IF offers additional favorable consequences over DCR on metabolic syndrome and its components, including non-alcoholic fatty liver disease (NAFLD) [2].

IF restricts meals to the daytime, which optimizes metabolic balance by overlapping the circadian rhythm [3]. The molecular processes that regulate body weight and energy metabolism are modified because of the diurnal pattern imposed by IF. Chrononutrition examines the relationship between mealtimes and the circadian cycle, as well as other physiological or behavioral processes that are intrinsic to living organisms [4]. The suprachiasmatic nucleus (SCN) coordinates the peripheral clocks from the intestine, pancreas, and liver, responds to food intake, and regulates intermediate metabolisms [5]. The SCN, which acts as the central conductor, ensures that the clocks in the periphery are synchronized, while feeding time, which acts as a zeitgeber, ensures that the clocks are precisely integrated. These rhythms permit endogenous processes to adapt effectively to environmental changes, such as the daily cycle of light and darkness and the cyclical availability of food [3,4,5]. During the dark phase, functional energy metabolism processes are at their maximum, while pathways for detoxification, motility, and environmental sensing attain their peaks during the day [6].

Dietary changes have a rapid and long-lasting effect on the gut microbiota [7]; however, these modifications are actually the result of alterations in microbial activity and genetic expression [8]. The relationship between diet and intestinal microbiota is bidirectional and complicated. Short-term dietary changes have an immediate impact on microbiota diversity, whereas long-term eating habits have a significant impact on composition [9]. As studies in humans and animals have revealed, intestinal dysbiosis may be caused by disruption to the molecular clock’s circadian rhythm due to environmental perturbation, such as dietary stressors. The structure or function of intestinal microbiota displays circadian patterns, and disruptions in the circadian rhythms of the host can have an effect on bacterial populations [10]. The gut microbiota inflects the interplay between meal timing and host circadian rhythms [6], actively orchestrates host diurnal rhythms through a variety of mechanisms, and acts both locally and remotely, such as the in the liver [6,11].

The human intestine contains a significant population of bacteria, fungi, and viruses. These organisms play essential roles in the breakdown of cellulose and in the provision of the substrate for gluconeogenesis and lipogenesis at the intestinal epithelium level [12]. Short-chain fatty acids (SCFAs), such as propionate, butyrate, and acetate, which are synthesized via fermentation of dietary fibers, exert a significant influence on hepatic function through various mechanisms [12]. Furthermore, the gut microbiota is involved in the fermentation of amino acids, in the conjugation of bile acids, in the synthesis of vitamins and in the control of the motor and immune systems of the gastrointestinal tract [13]. Metabolic health is determined by the diversity of genes associated with the gut microbiota and, consequently, the variety of microbial intestinal flora. Different microbiota compositions have been observed in relation to geographical location and general health [3,7,14]. Diet has a significant impact on microbiota composition, leading to the development of a stable microbial community known as an enterotype [7].

Animal studies’ findings suggest that IF can have positive effects on the microbiota by promoting diversity and cyclic activity, as well as modulating the abundance of specific bacterial species [11]. These effects might be particularly helpful in minimizing the effects of a high-fat diet on the microbiota and in lowering the number of obesogenic bacteria [15,16]. In mouse models lacking the circadian clock, IF induced the cyclic activity of the intestinal microbiota [11,15,17]. Additionally, IF was found to modulate the abundance of species that are characteristic of an unhealthy, obese microbiota [18]. In experimental animals with changes in the microbiota produced by a high-fat diet, IF led to an increase in protective bacteria belonging to the family Ruminococcaceae and a decrease in obesogenic flora (Lactococcus) [15,16].

Knowledge about the repercussions of fasting on the human gut microbiota is still insufficient. The modulator effect of IF on gut microbiota and the metabolic consequences of this interaction have not been completely well established and understood until now. Moreover, differences in gut microbiota and metabolic pathways between human and animal models have motivated us to synthesize the results provided by human studies. Our research aims to establish a framework of current information on the influence of IF on human microbiota and the possible connections to components of the metabolic syndrome. The potential crosstalk between fasting, gut microbiota, and metabolic syndrome has not been fully explored, and our research aims to contribute to this area of study. By examining the existing literature, we hope to identify gaps in knowledge and potential avenues for future research in this field.

## 2. Materials and Methods

This review conformed to the Preferred Reporting Items for Systematic Reviews and Meta-Analyses (PRISMA) Extension for Scoping Reviews (PRISMA-ScR) checklist [19].

### 2.1. Inclusion Criteria

Original research studies published in English addressing the topic of IF and gut microbiota comprised our review.

### 2.2. Data Sources and Search Terms

We searched the PubMed, Web of Science and Embase databases. We applied the search terms mentioned below:

(*‘intermittent fasting’ OR ‘intermittent energy restriction’ OR ‘periodic fasting’ OR ‘alternate day fasting’ OR ‘modified alternate-day fasting’ OR ‘modified alternate day fasting’ OR ‘time-restricted feeding’ OR ‘time restricted eating’ OR ‘whole day fasting’ OR ‘Ramadan’ OR ‘Ramadhan fasting’) AND (‘microbiota’ OR ‘microbiome’*).

### 2.3. Study Selection Process

The process of independent search, inclusion, and coding of studies by two researchers ensured the reliability and validity of the results. The involvement of a third senior researcher in resolving disagreements further strengthened the rigor of the study. Consensus ultimately resolved the remaining differences.

We looked at an initial sample of 985 studies (382 from PubMed/Medline, 305 Web of Science, and 298 from Embase) published until February 2023. We were left with 619 studies after eliminating duplicate titles. The remaining studies were then screened for relevance based on their titles and abstracts. We identified 111 studies that satisfied our selection requirements after we set aside reviews, reports, letter to editors, current opinions, and studies unrelated to our objectives. These studies subsequently underwent a review procedure to extract relevant data for our analysis. After selection, 30 studies that examined the relationship between fasting and gut microbiota were further analyzed. The PRISMA flowchart depicts the research selection process and the number of identified studies for each stage (Figure 1).

In total, 17 studies evaluated the effects of IF (six on time-restricted feeding (TRF); four alternate-day fasting (ADF); seven on Ramadan fasting), and seven on investigated prolonged fasting (four on the Buchinger diet; three on fasting-mimicking diets (FMD)). The Buchinger diet discontinues food intake over a variable period of time, generally 5–7 days; however, fasting periods do not alternate with those with normal caloric restriction as in the case of IF. In this review, we addressed the effects of IF; therefore, 17 studies were finally included in our analysis. The majority of these studies were interventional; only nine included a control group.

### 2.4. Quality Valuation

In conformity with recommendations for scoping reviews, we did not assess the quality or risk of bias of the included studies [20].

## 3. Results

Fasting is defined as the cessation of caloric intake for a variable duration, ranging from 12 h to 3 weeks [2]. In addition to religious fasting, various other forms exist, such as alternate-day fasting (ADF), the 5:2 diet, and time-restricted feeding (TRF) [4].

In this paper, we focused on the interactions of IF with gut microbiota composition and structure on the taxa level, as well as the metabolomic and metabolic interactions, to reveal the potential beneficial effects (Figure 2).

### 3.1. Time-Restricted Feeding (TRF)

Time-restricted feeding, or time-restricted eating, is a type of IF characterized by the absence of food intake for a period longer than 16 h, which may vary depending on the time of day. With feeding intervals ranging from 4 to 12 h, participants typically consume food in the morning–early afternoon, afternoon–early evening, or entirely in the evening. The duration of the eating window seemingly influences the efficacy of TRF on weight loss, with earlier or shorter windows providing more considerable benefits [21].

The objectives of the studies included in our review were: the evaluation of the influence of TRF on gut microbiota diversity [21,22,23,24]; the assessment of metabolic risk markers in relation to circadian rhythm and the gut microbiota [25]; the effect on lipid metabolism [23]; and the alterations of gut microbiota composition driven by nutrient intake [21,26]. A brief description of the TRF studies is given in Table 1.

The microbiota of individuals with obesity is different from that of normal-weight individuals. Obese individuals’ microbiota has been linked to a change in intestinal permeability, which permits bacteria to pass through the intestinal wall and cause inflammation throughout the body [3]. Patients with obesity can also be distinguished by the diversity of their bacterial composition. However, the definition of a “healthy” gut microbiome is still up for debate [27,28]. The obesogenic microbiota is characterized by lower diversity [29,30], altered abundance of Bacteroidetes phylum and Firmicutes phylum species [31], and changed composition (the increased presence of some species included in the Actinobacteria phylum) [32,33]. There are debates as to whether the microbiota changes that precede or accompany obesity are the ones that reflect a certain phenotype.

#### 3.1.1. Effects of TRF on Microbiota Diversity

Some of the included studies have examined the impact of TRF on microbial diversity in both healthy individuals and persons with obesity. In healthy individuals, Zeb et al. found microbial diversity significantly increased in the TRF group [26]. Conversely, Khan did not observe this effect [23]. Studies conducted on individuals with obesity found no significant differences in microbial diversity between the TRF group and the control group after 12 weeks [21,22]. When Xie et al. [24] investigated the impact of TRF on microbial diversity at different times of the day, they found that food intake restricted to the early part of the day (eTRF) contributed to a significant increase in microbial diversity (*p* = 0.048). However, neither mid-day TRF (mTRF) or the control group exhibited a significant increase in microbial diversity [24].

There seems to be a weaker TRF effect on microbiota diversity in the patients with obesity. However, when TRF was applied at different moments, particularly in the early part of the day, the stronger effect on increasing diversity seems to suggest that there is a connection with the circadian rhythm. Therefore, more research is needed to evaluate the effect of TRF on microbiota at different times of day in relation to BMI.

#### 3.1.2. Effects of TRF on Microbiota Community Composition

In Zeb’s study [26], the Bacteroidetes phylum was more abundant in the TRF group than the Firmicutes phylum, which was most abundant in the non-TRF group. This ratio demonstrated the beneficial effect of TRF on gut flora. Moreover, the changes in phylum level depended on participants’ nutritional state and gender [23]. In women who were overweight or with obesity, Firmicutes, Actinobacteria, and Bacteroidetes decreased, while Proteobacteria increased in response to IF. For patients with obesity, no changes were found in the numbers of Firmicutes and Bacteroidetes throughout the trial [22].

Several studies investigated the impact of TRF on bacterial species that produce butyrate. However, the studies reported conflicting results, with some showing an increase in the number of bacteria that produce butyrate and others showing no significant change in the relative abundances of such species. Zeb et al. [25] investigated bacterial species at different levels and demonstrated that TRF significantly changed bacterial species’ proportions, with the Ruminococcus family that produces butyrate being significantly higher in TRF samples.

The research findings indicate that a shift in gut microbiota represented a potential contributor to the positive effects of TRF, as SCFAs act as signals for satiety by activating certain proteins on the intestinal surface, namely GRP41 and GRP43, which are G-protein-coupled surface receptors [8,9,31,34,35]. When GRP41 is activated, enteroendocrine cells produce more peptide YY, which is a hormone that reduces the body’s capacity to absorb energy from food, leading to changes in peripheral glucose utilization [9,12,35]. SCFAs, along with other neurotransmitters, such as GABA, serotonin, and 5HI, which are produced during food metabolism under the influence of the intestinal flora, help regulate eating behavior by activating enteroendocrine cells and stimulating the synthesis of various intestinal hormones, such as GLP1, peptide YY, and cholecystokinin [36,37,38].

#### 3.1.3. Metabolomic Effects

The core circadian clock regulates biological processes based on a 24-h cycle, which is controlled by the transcription factors “Clock” (circadian locomotor output cycles kaput) and “Bmal1” (brain and muscle aryl hydrocarbon receptor nuclear translocator protein 1). These transcription factors initiate the transcription and translation of clock-controlled genes, including Period (Per) and Cryptochrome (Cry), which play a role in the feedback inhibition that prevents further transcription by Clock and Bmal1 [5]. The clock is regulated by fine-tuned mechanisms that modulate the activity of nuclear receptors, such as retinoic-acid-related orphan receptor alpha (Rora), reverse erythroblastosis virus alpha (Rev-erbα), and sirtuin 1 (Sirt1) [10].

Zeb et al. [25] found that TRF enhances circadian clock gene expression and improves metabolic regulation. The study observed that TRF intervention significantly increased the mRNA levels of the Bmal1 and Clock genes compared with the pre- and non-TRF groups. All members of the eTRF group displayed increases in the MESORs (the mean value based on the distribution of values across the cycles of the circadian rhythm) for Bmal1, Per2, and Sirt1 mRNA expression [24].

Sirt1 activation can regulate the circadian rhythm because its mRNA level is significantly elevated before and after fasting, as well as in the groups without TRF. The amplitude of all the subjects’ Sirt1 mRNA expression increased in the eTRF group [24]. The relative abundance of Prevotellaceae, Prevotella_9, and Bacteroidia was also associated with Sirt1 expression [25], which is known to have a substantial favorable connection with the Clock gene [39].

#### 3.1.4. The Relation between TRF-Induced Weight Loss and Microbiota

According to the nutritional status of the volunteers [23], the TRF had varying effects on the BMI. The participants with obesity had decreased BMIs at the conclusion of the study, while those with normal BMIs maintained their weight. A positive correlation was found between an increase in bacterial diversity and weight loss, while a decrease in bacterial diversity was noticed in women who gained weight during the trial. These results suggest that TRF may be more effective for obese individuals than for those of normal weight. Moreover, alterations in bacterial diversity may be relevant in the differential response to TRF among individuals with varying BMIs [23].

Another study included in our analysis did not find a similar relation. There was no correlation between weight loss induced by TRF, changes in fat mass, and blood pressure, and the diversity or abundance of microbiota [22].

In Xie’s study [24], despite similar changes in caloric intake in all TRF groups, body mass decreased only in the eTRF group as a result of decreases in both body fat percentage and fat mass. Moreover, the changes in the amplicon-sequence variants (ASVs), a measure of analyzing microbial communities, were not caused by changes in weight or diet but rather by shifts in meal timing [21]. The absence of significant correlations between ASVs and anthropometric or nutritional characteristics suggests that changes in ASVs were not attributable to alterations in diet or weight but rather to changes in mealtime schedules.

#### 3.1.5. Interactions with Metabolic Syndrome Components

The increased gut microbiota richness as a result of TRF [22,26] and the alteration of the proportion of Bacteroidia [26], Bifidobacterium and Lactobacillus [22] species were related to improvements in lipid profile. Microbial diversity was increased by TRF and was positively correlated with HDLc and Sirt1. Other studies observed that reduced bacterial diversity was associated with elevated blood lipids compared with a control cohort [40].

In addition, eTRF lowered TNF-α and IL-8 plasma concentrations in non-obese people, hence reducing inflammation [24]. Obesity is characterized by a low level of chronic inflammation and the development of non-alcoholic steatohepatitis. Thus, it was suggested to be a potential protective effect of eTRF against hepatic steatosis [24], which is in line with the findings of the vast majority of earlier investigations using NAFLD and liver steatosis animal models [41,42].

### 3.2. Ramadan Fasting

Ramadan is an annual period of religious fasting that involves abstaining from food consumption for 15–16 h per day for 30 days. This type of fasting has been related to various health benefits, including promoting blood sugar control and reducing insulin resistance. Studies have shown that Ramadan does not differ from other experimental TRF research, as there is no calorie restriction during Ramadan fasting [15]. However, the timing of fasting from dawn to dusk, which is a nocturnal-feeding regimen, may perturb human circadian rhythms [3,16]. All identified studies on the relationship between Ramadan fasting and gut microbiota were conducted on healthy men volunteers. The aim of the studies identified in our review was to describe the changes in the diversity and composition of the microbiota after the Ramadan fasting, the relation of these changes with weight modifications, the relation between dietary consumption and microbial taxa, and the effect on lipid profile serum levels. A summary of the studies of Ramadan effects on the microbiota is provided in Table 2.

#### 3.2.1. Effects of Ramadan Fasting on Microbiota Diversity

Depending on the evaluation method, the effects of Ramadan fasting on the microbiota diversity of healthy, normal-weight participants are contradictory. Operational taxonomical units (OTUs) data revealed a significant increase in microbial diversity during Ramadan [43]. Ozkul et al. [43] and Mindikoglu et al. [44] found no differences in α-diversity measurements following the Ramadan fast, whereas another study found a significant increase in α-diversity following the fast in a young cohort [45]. In the group of middle-aged individuals, however, this effect was not statistically significant [45]. The β-diversity analysis of revealed a significant difference between the pre- and post-Ramadan microbial community structure in Ozkul’s [43] and Su’s [45] studies, but not in Mindikoglu’s [44]. Following the cessation of fasting, the gut microbial community exhibited a strong tendency to return to baseline levels [45]. The resilience (resistance to environmental perturbations), stability (relative stability of the microbial community over time), functional redundancy (protection of functionality despite compositional changes), and recovery capacity of the healthy gut microbiota are distinctive features that can explain how microbiota withstands environmental changes [50,51].

One particular study described the effects of fasting in two groups: Chinese and Pakistani individuals [46]. It found that differences in α-diversity were driven by ethnicity rather than fasting, because the Pakistani group displayed much higher OTUs abundance indices than the Chinese group. The microbial community composition of the Chinese group changed very little after the fast, whereas the microbiota composition of the Pakistani participants showed significant divergence with little overlap. As a result, the authors hypothesized that dietary modifications can significantly determine how ethnicity and Ramadan fasting affect the gut flora [46]. As this was the only identified study focusing on ethnic differences, the results should not be generalized, and more research on this area is needed.

#### 3.2.2. Effects of Ramadan on Microbiota Community Composition

The increase in Bacteroidetes and Firmicutes abundance was a common finding [43,47,52], and the Firmicutes/Bacteroidetes ratio, an indicator of the microbiota associated with obesity, remained increased after Ramadan fasting [43]. Furthermore, changes in the microbiota seemed to be related to ethnicity. Thus, Proteobacteria were enriched after fasting, possibly due to a significantly higher abundance of Klebsiella, whereas Bacteroidetes were more prevalent among Chinese individuals before fasting. Firmicutes were significantly more abundant among Pakistani subjects prior to fasting, while Bacteroidetes were significantly more prevalent after fasting [46].

The compositional changes in gut microbiota following the end of the Ramadan fasting are highlighted by the results of the studies included in our analysis. Ramadan fasting significantly increased some taxa (Bacteroides, Butyricicoccus, Eubacterium, Faecalibacterium, and Roseburia [43]), which are altered in obesity. However, the exact relationship between these genera and obesity is still not fully understood [53,54]. These compositional changes primarily involve an increase in butyrate-producing bacteria, which could be interpreted as a trend toward a balanced gut microbiota [43,48]. After Ramadan fasting, the total bacterial count and number of Bifidobacterium and Lactobacillus species decreased significantly in fecal samples from nine healthy volunteers [48]. Understanding the underlying mechanisms of these changes and their implications for human health requires additional research.

Another significant effect of IF was a reversible increase in the number of Lachnospiraceae family members [45]. Together with other Clostridiales, the Lachnospiraceae are the primary cause of butyrogenesis in the human intestine. Butyric acid, a significant source of energy for the epithelial compartment, promotes metabolic benefits via gut–brain neuronal pathways [36,37,38].

#### 3.2.3. The Relation between Ramadan Fasting-Induced Weight Loss and Microbiota

A meta-analysis demonstrated a significant decrease in fat percentage and weight in individuals with overweight or obesity after the Ramadan fast [55]. Akkermansia muciniphila, a mucin-degrading bacterium whose relative abundance is inversely correlated with body weight, became more abundant after Ramadan fasting [48]. Bacteroides fragilis group members increased in abundance after the fasting period [48]. The decrease in BMI after Ramadan fasting (*p* = 0.05) was associated with an increase in Bacteroides in the gut microbiota [47]. The Bacteroides genus is a component of healthy gut microbiota that is highly capable of metabolizing complex polysaccharides and fatty acids [56] and was also found to be increased in adolescents following an obesity treatment program involving a reduction in energy intake [57]. Changes in the abundance of Bacteroides and Akkermansia muciniphila after Ramadan fasting may be related to weight loss.

On the other hand, Su [45] observed that Bacteriodes phylum OTUs, specifically those belonging to the Prevotellaceae family, decreased during the fast. Because there was no correlation between the composition of the gut microbiota and BMI prior to fasting, these changes appeared to be more related to Ramadan fasting than to the reduction in BMI overall [45]. Future research is required to determine whether these microbiota changes are a result of fasting or whether they are caused by weight loss.

#### 3.2.4. Metabolomic Effects and Interactions with Metabolic Syndrome Components

In a study on Chinese and Pakistani participants, changes in metabolite composition and the content of feces were registered after fasting in spite of ethnic particularities [49]. According to these results, some fecal metabolites levels (L-histidine, cordycepin, and lycofawcine) were considerably higher in the Chinese participants [49]. In mice under metabolic stress, cordycepin had protective effects against liver injury [58].

In addition, another study [44] that investigated the effects of Ramadan fasting on the serum proteome and key regulatory proteins in healthy individuals confirmed its protective role on hepatic steatosis. The authors [44] noted the upregulation of PLIN4 (perilipin 4, a protein involved in the storage of lipids within cells) and PKM GP (pyruvate kinase M1/2, an enzyme involved in glycolysis) expression in the absence of calorie restriction or substantial weight loss, which points to the possibility that 30-day IF may be highly effective in managing metabolic syndrome. NR1D1 levels (nuclear receptor subfamily 1 group D member 1, known as Rev-erbα), a circadian clock component, increased significantly after the Ramadan fasting period, which may lead to less fat accumulation in the liver and adipose tissues [44]. Furthermore, the ASGR2 GP level (asialoglycoprotein receptor) increased, which is an essential receptor for the liver’s ability to clear away apoptotic cell debris and regulate the immune system [44]. Overall, these results suggest that IF can be used as an additional form of care for those with metabolic syndrome and NAFLD.

### 3.3. Alternate-Day Fasting and 5:2 Diet

These types of IF are based on the idea that white adipose tissue lipolysis and the use of free fatty acids, ketone bodies, and glycerol as energy sources are triggered when food intake is absent for more than 12 h, which happens when the liver’s glycogen stores are depleted [3]. ADF is a type of IF in which days of severe caloric restriction are alternated with days of unrestricted eating. It comes in two forms: zero-calorie alternate-day fasting (energy intake is restricted to zero calories during the fasting days) and modified ADF (energy intake during fasting days is provided by a single meal that accounts for 20–30% of normal consumption) [59,60]. The 5:2 diet consists of two (consecutive or not) days per week of severe caloric restriction and five days of unrestricted consumption [59,60]. A succinct presentation of ADF and 5:2 studies is depicted in Table 3.

#### 3.3.1. Effects on Microbiota Diversity

ADF for three months resulted in a shift in overall microbiota community structure and richness (α and β-diversity) in participants with overweight and obesity [64]. One pilot study found that ADF increased bacteria variety and that the diversity of the microbiota was inversely related to the amount of leptin in the blood [61]. However, there were not significant differences in α- and β-diversity between DCR and the 5:2 diet. The average diversity of gut microbiota was minimally affected, suggesting that short-term IF regimens have a minimal impact on the gut microbiota of volunteers with obesity or overweight [62,63]. Thus, gut microbiota demonstrated remarkable resilience despite significant weight loss and cardio-metabolic improvement. Moreover, the correlation between baseline α-diversity and the loss in body fat suggested that the diversity of bacterial species at the start of the intervention had a significant role in the success of weight-loss interventions [62].

#### 3.3.2. Effects on Microbiota Community Composition

Cignarella found that Faecalibacterium and Blautia, two genera of bacteria within the phylum Firmicutes, increased their abundance in the IF arm [61]. Two members of Firmicutes phylum, Ruminococcaceae at the family level, and Roseburia at the genus level exhibited increases in abundance subsequent to the 5:2 diet [63]. The gut microbiota composition for both groups in Mohr’s study [62] was dominated by Firmicutes, followed by Actinobacteriota, Bacteroidota, Proteobacteria, and Desulfobacterota. There were no significant differences in the abundances of taxa above the genus level from baseline to the 4th week, nor a significant interaction effect on the Firmicutes/Bacteroidota ratio. Ruminococcaceae Incertae Sedis increased in both interventional groups [62]. Incertae Sedis is an unclassified genus belonging to the Ruminococceacae family that plays a crucial role in gut health due to its ability to produce butyrate and other SCFAs [62].

In both the DCR and IF intervention groups of the DRIFT2 study [64], the types and abundances of some microbes changed significantly from baseline to three months: Bacteroides and Alistipes abundance increased and Collinsella decreased. Bacteroides increased with hypocaloric weight-loss diets [57], and Alistipes abundance was related to sustained weight loss following DCR [65]. While there was an augmentation in Akkermansia in the IF intervention group, no change was noticed in the DCR group. The DRIFT2 study suggests that this difference could be due to the particular effects of more prolonged periods of fasting, although further research is needed [64].

#### 3.3.3. Metabolomic Effects and Interactions with Metabolic Syndrome Components

IF induced similar alterations in human leptin levels and gut microbiota to those in mice with experimental autoimmune encephalomyelitis [61]. The relation between Faecalibacterium and serum adiponectin levels indicated a consistent pattern of metabolic changes and gut microbiota alterations in metabolic syndrome patients undergoing IF [61].

IF improved gut-related metabolites, decreased lipopolysaccharides, and increased the relative abundance of SCFA-producing species [63]. Thus, IF improved intestinal barrier function and decreased the host’s systemic inflammation [63].

Serine, citrate, glucuronic acid, and TMAO (trimethylamine-N-oxide) had variations in the plasma concentration during the 5:2 diet [62]. The first three products have beneficial roles in metabolism (energy metabolism for the first two, and detoxifying metabolic pathways for the latter) [62]. TMAO, a compound found in high-protein foods, has been linked to heart disease, and appears to be diet-dependent [62].

## 4. Conclusions

Despite conflicting results and limitations, such as the small number of participants or the lack of a control group, the majority of studies have positive findings, highlighting the significance of this area of research and suggesting that IF may modulate metabolic health beyond the effects of caloric restriction and weight changes.

The results suggest that IF affects the microbiota in terms of abundance and composition. Pre-existing nutritional weight status and previous diet pattern seem to influence the alterations in gut microbiota during IF, but the short-term compositional gut microbiota changes return to the initial state upon cessation of fasting. The timing of the fasting period seems to be important in the gut microbiota response. Another important finding is that IF increases the abundance of butyrate-producing bacteria species, thus promoting metabolic benefits.

More research is needed to determine how IF affects the gut microbiota of normal-weight people compared with those who are obese or overweight. Another important point is to clarify the specific role of diet caloric restriction, fasting, and synchronization with the circadian rhythm in shaping the gut microbiota. Furthermore, future studies should investigate, over longer periods of time, the effects of IF on gut microbiota composition. This will provide a better understanding of the potential benefits and limitations of this dietary approach. Moreover, additional research is necessary to determine what occurs when IF is combined with probiotics or with other types of diets, such as DASH or the Mediterranean diet.

## Figures and Tables

**Figure 1 nutrients-15-02095-f001:**
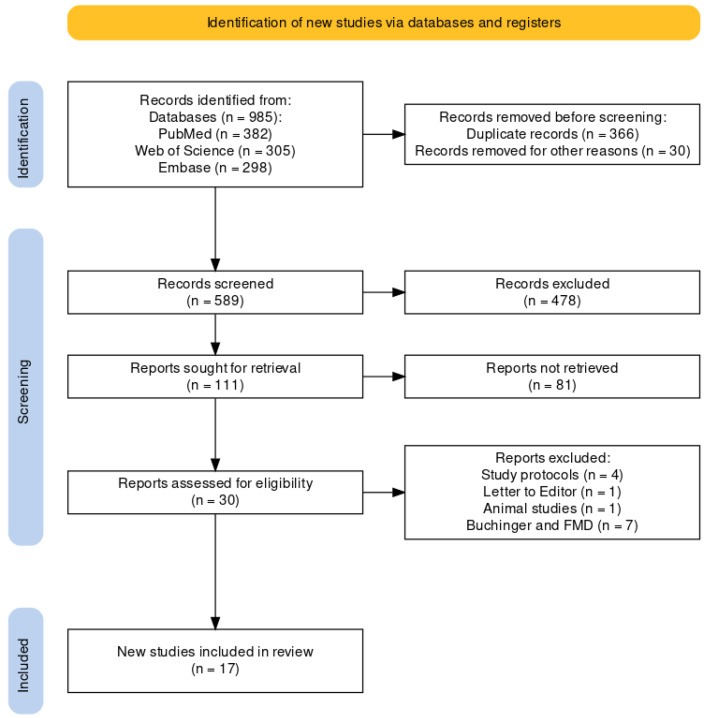
PRISMA overflow and number of papers included.

**Figure 2 nutrients-15-02095-f002:**
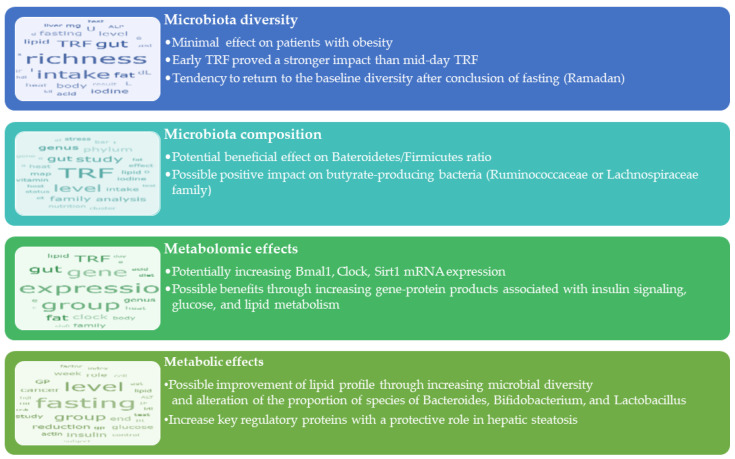
Potential beneficial effects of IF on gut microbiota and metabolism.

**Table 1 nutrients-15-02095-t001:** Brief description of the TRF studies.

TRF	Study Type	Duration	Study Population	Control Group	Number of Participants	Results
Ferrocino et al. [21]	Real-life study	12 weeks	BMI 30–45 kg/m^2^	TRF (*n* = 25) non TRF (*n* = 24)	49	No differences in α and β diversity or gut microbiota composition.
						Increase in Lachnospiraceae, Parasutterella, and Romboutsia in the TRF group.
Gabel et al. [22]	Pilot study	12 weeks	BMI ≥ 30 kg/m^2^	no	14	Regarding gut microbiota diversity, the abundance of Firmicutes, Bacteroidetes, or other phyla remained unchanged.
Khan et al. [23]	Clinical trial	26 days	Healthy volunteers (normal weight or obesity)	no	45	Alteration of the gut metagenome after IF.
						Increased bacterial α diversity.
						Lactobacillus and Bifidobacterium increased.
Xie et al. [24]	RCT	5 weeks	Healthy individuals without obesity	eTRF (*n* = 28)	82	No significant alteration in the gut microbiota diversity or overall composition.
				mTRF (*n* = 26)		eTRF > mTRF in improving insulin sensitivity.
				control group (*n* = 28)		Only eTRF improved fasting glycemia, decreased adiposity, ameliorated inflammation, and increased gut microbiota diversity.
Zeb et al. [25]	RCT	25 days	Healthy men	TRF (*n* = 56) non TRF (*n* = 24)	80	Enrichment of Prevotellaceae and Bacteroideaceae in the IF group.
						Activation of sirtuin-1 was positively related with gut microbiota richness (TRF).
Zeb et al. [26]	RCT	12 weeks	Healthy men	TRF (*n* = 15)	30	Increased Prevotella_9, Faecalibacterium, and Dialister in TRF.
				non-TRF (*n* = 15)		Bacteroidetes were the most abundant in the TRF group, while in the non-TRF group, Firmicutes was the prevailing phylum

**Table 2 nutrients-15-02095-t002:** Brief description of the Ramadan studies.

Study	Study Type	Control Group	Number of Participants	Results
Ozkul et al. [43]	Pilot study	No	9	Microbial richness increased. No significant difference in terms of α-diversity. Significant differences between baseline and after Ramadan in microbial community structure: Firmicutes/Bacteroidetes ratio remained increased; however, a higher abundance of Bacteroidetes was noticed when comparing with baseline. Butyricicoccus, Bacteroides, Faecalibacterium, Roseburia, Allobaculum, Eubacterium, Dialister, Erysipelotrichi, and Akkermansia were enriched.
Mindikoglu et al. [44]	Clinical trial	No	14	Bacterial richness and diversity did not change significantly.
				Increase in gene-protein products associated with glucose and lipid metabolism and insulin signaling (perilipin 4, pyruvate kinase M1/2).
Su et al. [45]	Cohort	30 young men and 37 middle-aged men (10 control, 27 Ramadan)	67	Increased microbiome diversity associated with upregulation of the Clostridiales-order-derived Lachnospiraceae and Ruminococcaceae.
				Decreased abundance of the Prevotellaceae family.
				Microbiome composition returned to baseline when IF finished.
Ali et al. [46]	Cohort	No	34 (16 Chinese and 18 Pakistani adults)	α-diversity significantly altered among Chinese subjects.
				IF could affect β-diversity in both populations.
				Increased relative abundance of Proteobacteria (combined groups analyses).
				Decreased relative abundance of Firmicutes (Pakistani group).
				Increased relative abundance of Bacteroidetes phylum (Chinese group).
				Decreased relative abundance of Bacteroidetes phylum (Pakistani group).
Mohammadzadeh et al. [47]	Cross-sectional	No	30	Bacteroides and Firmicutes increased.
				Bacteroides increased in both sexes.
				Firmicutes increased only in women.
				Increased butyrate levels after IF.
Özkul et al. [48]	Pilot study	No	9	Increased abundance of A. muciniphila and B. fragilis groups.
Chen et al. [49]	Cohort	No	34 (16 Chinese and 18 Pakistani adults)	L-histidine, lycofawcine, and cordycepin levels were higher after IF in the Chinese group.
				Brucine increased in the Pakistani group.
				Bacterial taxa were correlated with specific metabolites unique to each ethnic group.

**Table 3 nutrients-15-02095-t003:** Brief description of the effects of ADF and the 5:2 diet on gut microbiota.

Study	Study Type	Duration	Study Population	Control Group	Number of Participants	Results
Cignarella et al. [61]	Pilot study/ADF	15 days	Metabolic syndrome patients	8 ad libitum	16	IF confers protection through gut microbiota changes in metabolic syndrome patients.
				8 IF		
Mohr et al. [62]	RCT/5:2	5 weeks (4 weeks intervention)	Sedentary volunteers with overweight/obesity	1-day fasting (*n* = 10)	20	β-diversity was altered. Increased abundance of Ruminococcaceae Incertae Sedis in both groups. Increased abundance of Eubacterium fissicatena in the 1-day fasting group.
				2-day fasting (*n* = 10)		
Guo et al. [63]	RCT/5:2	8 weeks	Metabolic syndrome patients	21 IF	39	Increased production of SCFAs.
				18 controls		Decreased levels of lipopolysaccharides.
Stanislawski et al. [64]	RCT/ADF	12 months	Overweight/obese participants	25 DCR	71	Increased number of bacterial taxa were associated with weight loss and reduction in waist circumference.
				34 IF		

## Data Availability

No new data were created or analyzed in this study. Data sharing is not applicable to this article.

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
