# Peer review of "A Scoping Review of the Relationship between Intermittent Fasting and the Human Gut Microbiota: Current Knowledge and Future Directions"

_nutrients, 2023, doi:10.3390/nu15092095_

Round 1

Reviewer 1 Report

The present study give insight in relationship between intermittent fasting and the human gut microbiota. The manuscript was well addressed, I would like to suggest minor revisions.

Minor points:

Line 52: Gut microbiota is not often abbreviated as GM, I would like to suggest the author to use full name in the manuscript.

Line 77-78: Lack of connection here between gut microbiota and intermittent fasting, the author may add current progresses of relevant studies, and then point the gap.  

Table 1: zeb et al. (21). Other than “Enrichment of GM (Prevotellaceae and

Bacteroideaceae) in the IF group”, I would like to suggest “Enrichment of Prevotellaceae and Bacteroideaceae in the IF group”.

zeb et al. (22). It may cause misunderstand when mention” Bacteroidetes and

Firmicutes were the most abundant in TRF group” please double check this reference and rephrase the sentence.

Table 2: the author may check the reference and give more information on enrichment of specific taxa.

In addition, I would like to suggest the author to generate a figure as and overview of effect of IF on gut microbiota (either through enrichment of specific taxa or metabolism and etc)

Author Response

Dear reviewer, 

Please find attached our response.

Reviewer 2 Report

This review provides a timely discussion of the relationship between intermittent fasting and microbiota in the human gut. Owing to the widespread interest in intermittent fasting by researchers, clinicians, and the general public, this topic should be of high interest to readers of the journal. In particular, this study highlights that there is still much to be learned in this field. The study is presented in a straightforward manner, with little interpretation. I do not have any problems with the scientific validity of the study, and the study follows the format for a scoping review. However, there are frequent problems with presentation in the English language, and this needs to be improved. For instance, there are issues with the use of articles such as “the” throughout the manuscript, and word choice can be improved at many instances. I have highlighted some of the issues below in terms of major and minor comments, but this does not fully address the issues that are present. To help resolve these issues, I strongly suggest that the manuscript undergo a careful edit by an experienced scientific writer whose native language is English.

Major comments:

1.      Awkward construction on line 112: “thus remaining 30 studies”

2.      Awkward construction on lines 117-118: “Buchinger diet withdraw food intake”

3.      Awkward construction on line 231: “Other study didn’t find”

4.      On lines 290-292, you mention how their were ethnicity-dependent responses to Ramadan fasting. Is their any evidence that types of intermittent fasting affect other ethnicities differently? If so, this would negate generalizations that result from studies focusing on a single ethnicity.

5.      Awkward construction on line 340: “cordycepin proved protective effects…”

Minor comments:

1.      The first line of the abstract (line 11) should be re-worded as “Intermittent fasting (IF) has been promoted…”

2.      Line 15: The word “liaisons” typically refers to communication between people or organizations and should not be used to describe physiological relationships. I suggest using the word “relationship” here, especially since this is the same word used in the title.

3.      Comma at end of line 92 should not be there, as it is placed between the subject and verb.

4.      Lines 119, 231, 371: Contractions such as “don’t” should not be used in formal writing.

5.      Line 256: This should be re-worded as “This type of fasting has been related to…”

6.      Throughout the manuscript, “also” is commonly used as a transcription word at the beginning of a sentence. However, “also” is considered a less formal way to start a sentence, thus its use should be avoided in formal, academic papers. Better alternatives include moreover, furthermore, and in addition.

This review provides a timely discussion of the relationship between intermittent fasting and microbiota in the human gut. Owing to the widespread interest in intermittent fasting by researchers, clinicians, and the general public, this topic should be of high interest to readers of the journal. In particular, this study highlights that there is still much to be learned in this field. The study is presented in a straightforward manner, with little interpretation. I do not have any problems with the scientific validity of the study, and the study follows the format for a scoping review. However, there are frequent problems with presentation in the English language, and this needs to be improved before the manuscript is publication-ready. For instance, there are issues with the use of articles such as “the” throughout the manuscript, and word choice can be improved at many instances. I have highlighted some of the issues below in terms of major and minor comments, but this does not fully address the issues that are present. To help resolve these issues, I strongly suggest that the manuscript undergo a careful edit by an experienced scientific writer whose native language is English.

Author Response

Dear reviewer,

Please find attached our reply. We also attached the English Editing Certificate for our manuscript.

Reviewer 3 Report

This review article entitled, “A Scoping Review of the Relationship Between Intermittent Fasting and the Human Gut Microbiota: Current Knowledge and Future Directions” was designed, organized, and written well.  It will be of interest to a large audience as it has a focus in 2 major areas including fasting and gut health.  I do not have any minor or major edits to suggest for this version prior to publication.

Author Response

Dear reviewer,

Thank you very much for your effort to read our manuscript and we are very grateful for your appreciations. 

Sincerely,

Otilia Niță,
